Models for the directed evolution of bacterial allelopathy: bacteriophage lysins

Bull James J. 1 2 3
Crandall Cameron 4
Rodriguez Anna 4
Krone Stephen M. 5 6 krone@uidaho.edu
1 The Institute for Cellular and Molecular Biology, University of Texas , Austin, TX , USA
2 Department of Integrative Biology, University of Texas , Austin, TX , USA
3 Center for Computational Biology and Bioinformatics, University of Texas , Austin, TX , USA
4 Department of Biological Sciences, University of Idaho , Moscow, ID , USA
5 Department of Mathematics, University of Idaho , Moscow, ID , USA
6 Institute for Bioinformatics and Evolutionary Studies, University of Idaho , Moscow, ID , USA
Dahlem Markus
Electronic publication date: 2015 Apr 9
Publication date: 2015
Volume: 3
Electronic Location ID: e879
Received 2015 Jan 5; Accepted 2015 Mar 16
Copyright: © 2015 Bull et al.
Copyright year: 2015
Copyright holder: Bull et al.
License: This is an open access article distributed under the terms of the Creative Commons Attribution License, which permits unrestricted use, distribution, reproduction and adaptation in any medium and for any purpose provided that it is properly attributed. For attribution, the original author(s), title, publication source (PeerJ) and either DOI or URL of the article must be cited.
License URL: https://creativecommons.org/licenses/by/4.0/

Keywords: Directed evolution, Lysin, Antibiotic

Funding: NIH 57756 NSF UBM DMS-1029485 JJ Bull was supported by NIH 57756 and the University of Texas Miescher Regents Professorship. Cameron Crandall and Anna Rodriguez were supported by NSF UBM DMS-1029485 (to Stephen Krone). The funders had no role in study design, data collection and analysis, decision to publish, or preparation of the manuscript.

==============================
Microbes produce a variety of compounds that are used to kill or suppress other species. Traditional antibiotics have their origins in these natural products, as do many types of compounds being pursued today in the quest for new antibacterial drugs. When a potential toxin can be encoded by and exported from a species that is not harmed, the opportunity exists to use directed evolution to improve the toxin’s ability to kill other species—allelopathy. In contrast to the typical application of directed evolution, this case requires the co-culture of at least two species or strains, a host that is unharmed by the toxin plus the intended target of the toxin. We develop mathematical and computational models of this directed evolution process. Two contexts are considered, one with the toxin encoded on a plasmid and the other with the toxin encoded in a phage. The plasmid system appears to be more promising than the phage system. Crucial to both designs is the ability to co-culture two species/strains (host and target) such that the host is greatly outgrown by the target species except when the target species is killed. The results suggest that, if these initial conditions can be satisfied, directed evolution is feasible for the plasmid-based system. Screening with a plasmid-based system may also enable rapid improvement of a toxin.

Introduction

Motivated by a mounting tide of drug resistant bacteria, the search for new antibacterial agents is embracing technologies that lie outside traditional bounds. One promising source of compounds is the lysins encoded by bacteriophages (Nelson, Loomis & Fischetti, 2001; Fischetti, 2003; Fischetti, 2008; Fischetti, 2010; Fischetti, 2011; Borysowski, Weber-Dabrowska & Górski, 2006; Loeffler, Nelson & Fischetti, 2001; Pastagia et al., 2013; Chan & Abedon, 2014). These enzymes degrade the bacterial cell wall, the peptidogylcan layer essential for cell survival against the osmotic gradient that otherwise threatens to rupture the cell. Phage lysins are ubiquitous, at least among phages with large genomes, so there is a potentially unlimited source of new ones, and they are also highly evolvable. Across diverse phages, the enzymatic basis of lysis is known to lie within a few protein-degrading activities (McGowan et al., 2012), and evidence from some phages suggests that lysins may have broad host ranges and not easily succumb to bacterial resistance evolution.

Yet from an evolutionary perspective, native lysins are expected to have some drawbacks as antibacterials in a clinical setting (Fischetti, 2011). (1) Lysins have evolved to work from inside the cell, where lysin concentration is high late in the infection cycle. In contrast, the clinical setting requires that they work from outside the cell and possibly at low concentrations. (2) Because phages are selected to avoid harming future hosts prior to infection under conditions of normal growth, lysins—which are released into the environment when the cell ruptures at the end of the phage infection cycle—have evolved to avoid killing cells from outside. These limitations are not necessarily insurmountable, but an obvious inference from them is that native lysins will have limits to their efficacy, limits that reduce their applications and commercial success, if only by requiring high doses.

Engineering offers one approach to lysin improvement. For example, a common structure of lysins found in phages of Gram positive bacteria is a binding domain with a separate enzymatic domain, allowing for easy retargeting of the enzyme domain by coupling it with alternative binding domains (Fenton et al., 2010; Pastagia et al., 2013). Structural analysis of active sites could potentially suggest ways of redesigning lysins to improve catalytic activity (cf., McGowan et al., 2012), and indeed, rational protein design of point mutations has been successful in improving activity (Díez-Martínez et al., 2013). Alternative to engineering is screening, perhaps of populations subjected to mutagenesis, an approach that has also led to improved activity or thermostability (Cheng & Fischetti, 2007; Heselpoth & Nelson, 2012).

Yet another approach for improving molecules is directed evolution: a large library or population of the lysin gene of interest is evolved for greater function (Farinas, Bulter & Arnold, 2001). The benefit of directed evolution over engineering is that it requires no prior understanding of the molecular basis of function. It thus finds solutions that engineering does not, at least until the knowledge base underlying the engineering improves substantially. The main difficulty in using directed evolution to improve agents that kill is that directed evolution relies on enhanced survival or reproduction. Agents that die with their hosts are not among the survivors, so better lysins are not selected unless they are coupled with their own survival.

Directed evolution of lysins thus requires that the lysin be encoded by a genome that benefits when the lysin kills the intended target. There are two feasible contexts for this solution. In one, the lysin is encoded by and exported from one species to kill a competitor of the species encoding the lysin, a phenomenon known in ecology as allelopathy. In a second context, the lysin is encoded on a phage that infects one species growing in competition with a second species; as in the first context, the phage-encoded lysin must kill the competitor, indirectly benefitting the host on which the phage depends. Both contexts are intuitively feasible, but on further inspection, the efficacy of directed evolution is sensitive to many quantitative details about the competition. The purpose of this paper is to use mathematical models to explore the properties of a directed evolution protocol that might render the selection feasible.

As inspiration, the directed evolution of allelopathy was demonstrated by Chao & Levin (1981) using colicins. That study used an intact bacterial toxin system (a bacteriocin or colicin) encoded on a plasmid and identified conditions under which cells carrying the colicin plasmid increased over sensitive cells lacking the plasmid. The main conclusion from that study was the requirement of spatially structured growth for the plasmid to increase when rare. Their study used a fully functional and lethal toxin system that needed no fine tuning; furthermore, the plasmid encoded both the toxin and a defense against the toxin, enabling the selection to be done within a single bacterial strain. Our purpose here is to use mathematical models and provide the foundation for improving an imperfect toxin system against all individuals of a target species, where no defense by that species is possible. We identify properties of a protocol that should enable such evolution and suggest difficulties with some designs that might appear intuitively feasible. The directed evolution of allelopathic toxins presents enough conceptual difficulties to justify the use of models.

General approach

The goal is to evolve improved killing of a species C. The process begins with a gene-encoded toxin that is only moderately effective at killing, and this gene will be the target of the evolution. (We use ‘toxin’ interchangeably with ‘lysin,’ recognizing that our models apply to more types of toxins than lysins.) Basic evolutionary theory indicates that, if selection is to improve the toxin (gene), there needs to be a benefit to mutant genes that kill C at a rate above the baseline exhibited by non-mutant toxin. As it is not feasible to select mutants with reproductive success lower than the mean, the toxin needs to be encoded in a genome that experiences improved reproductive success when C is killed.

An obvious context for this evolution is to encode the toxin on a plasmid (or in the genome) of a host species B (e.g., a bacterium); we refer to this below as the ‘plasmid’ model. It is obvious that two conditions must be satisfied for this protocol to work: (i) B is not harmed by the toxin. (ii) Growth of B is reduced by C when both species are cultured together—that is, C is a competitor of B. In theory, if a mutant toxin harbored by B reduces competition from strain C, that mutant should have an excess number of descendants and increase in frequency. Perhaps less obvious is (iii) the environment must impose some form of spatial structure: individuals of B that suppress competition from C must directly realize the benefit of reduced competition (Chao & Levin, 1981). This cannot be achieved with large populations grown in liquid, as any resources freed from dead individuals of C could be used by any individuals of B, not just those causing the enhanced killing.

There are various in vitro environments that might be used for this directed evolution. One is agar plates, both species being mixed, spread on the plate, and allowed to grow until resources are exhausted. After growth on a first plate, cells of species B would be selectively harvested, diluted and then replated in competition with new C. A similar type of selection, with perhaps greater manipulability, is emulsion droplets (Lu & Ellington, 2013). For our application, droplets of constant volume are inoculated with both types of cells, grown to saturation within the droplet and then harvested en masse for species B. The recovered B would then be emulsified again with fresh C and the growth phase repeated. We model both of these designs below.

Finally, an alternative approach to encoding the toxin on a plasmid is to encode the toxin gene in a phage that infects B but not C. Enhanced growth of B would translate to increased phage production. In this case, phage would be harvested at the end of each cycle, re-pooled, and then emulsified or plated with fresh B and C hosts. This design is modeled here as well.

Plasmid model 1: emulsion droplets

Single founders: evolution depends on relative advantage

We suppose that a large population of emulsion droplets is created, each droplet constituted with one or more individuals of B and one or more individuals of C. Within each droplet, B and C individuals grow in competition to droplet capacity, and the B individuals within all droplets are harvested (excluding any B that failed to be emulsified at the outset).

The dynamics of growth competition between B and C will often be complicated, but considerable insight can be gained by focusing on the competition endpoints after one cycle of growth. To address evolution, we consider two types of cells: B′ (the mutant) and B (the common type). After each incubation cycle, the resulting B′ and B cells are collected together without distinction and subjected to further selection. For now, we assume that any droplet gets only B or B′ cells, not both, along with competing C cells. Other droplet compositions will be considered later.

Table 1 defines notation for the derivations that follow.

Table 1 Notation.

Symbol	Definition	
h	The average number of B cells harvested from a droplet populated with C and only B (no B′)	
h′	The average number of B′-type cells harvested from a droplet populated with C and at least one B′	
pg	Frequency of B′ at end of cycle g	
1 − pg	Frequency of B at end of cycle g	
Bg	Number of B individuals at end of cycle g	
Bg′	Number of B′ individuals at end of cycle g	
q	At cycle outset, the average number of B + B′ cells inoculated per droplet	

An interesting consequence of this process is that evolution of a new toxin depends ultimately on the numbers of B (=h) and B′ (=h′) harvested from each type of droplet (see Table 1). The dynamics of C within a droplet of course affect these numbers, but selection carries forward only B and B′ cells, so any B individuals that leave more than the average number of descendants will increase.

With p0 indicating the initial frequency of B′ and pg (g ≥ 1) being the frequency at the end of cycle g (and the start of cycle g + 1) of B′ among all ‘B-types’ (B′ + B) in the population, the evolutionary dynamics has a simple form: (1) p11−p1=h′hp01−p0

for the first cycle, and (2) pg1−pg=h′hgp01−p0

after g cycles. Thus the relevant measure of mutant fitness (W) in this protocol is simply (3) W=h′h,

the number of B′ progeny harvested per droplet over the number of B progeny harvested per droplet.

Inclusion of mixed droplets does not profoundly change the result

The above equations assume an ideal initialization, each droplet containing just B or B′ (in addition to C). We now consider the effects of Poisson sampling and the resultant mixed droplets. We are interested in the ratios pg/(1 − pg), g ≥ 1, where p0/(1 − p0) is the initial ratio of B′ cells to B cells.

If droplets are initialized to have (Poisson average) q > 0 B-type cells, the ‘Appendix’ shows that (4) pk+11−pk+1=h′hpk1−pk⋅2+q2+q+qpkh′h−1.

This result is similar to that in (2) except for a correction term. When B′ is rare, pk is small and (4) is approximately the same as (1), regardless of q.

Given relatively limited information about fitnesses, this recursion can be used to make optimal choices for q and the number of incubation cycles (see below).

Implications for experimental design: growth conditions are critical

The rate of progress thus depends heavily on the relative superiority of the mutant to the common type, measured in the number of progeny per droplet. That quantity depends both on the protocol and on the phenotypic effect of the mutations available to the lysin. The latter lies in unknown biology of the lysin, but the former is under control of the experimenter. Our analysis allows us to make three points about how the design is critical to evolutionary progress.

1. The baseline number of B progeny harvested per droplet (h) should be low; a low h ensures a small denominator in the ratio h′/h, on which selection acts. Minimally, the competitor C must outgrow B in the absence of toxin, but ideally, C should even outgrow B with the initial toxin that is being evolved for improvement.

2. B-types must be able to produce progressively more progeny as more and more of C is killed or inhibited. Enhanced progeny production leads to a progressively higher numerator in the ratio h′/h as the toxin becomes more effective.

3. The dynamic range for growth of B-types should be large. The droplet volume should permit at least 10—ideally 100 or more—final bacteria. Droplet volume sets the limit on h′. A small droplet volume may enhance toxin efficacy, but it also limits the dynamic range.

These conditions define the foundation for a selection protocol. Items (1) and (3) can be established for a system even in the absence of toxin. Even condition (2) can be partly established in the absence of toxin by ensuring that B grows when C is absent.

It is also evident from these points that directed evolution of a toxin is vastly easier when an antidote can be encoded on the same plasmid as the toxin (as is true of naturally occurring bacteriocins). The selection can then be performed within a single species—the target species—between individuals with and without the plasmid. Since only a single species is involved, conditions (1) and (2) are alleviated because an appropriate growth environment for the selection is merely that in which the target species grows. Likewise, if individuals of C that are fully resistant to the toxin can be isolated, and if the toxin is exported from those types without harm to them, the selection can also be conducted within a single species. The broader the host range and mechanism of action, the greater the challenge to confining the protocol to a single species. Other considerations about baseline h and dynamic range apply regardless of the number of species.

Whether a directed evolution experiment can be successful depends not only on the rate of progress per cycle but also on the number of cycles. Directed evolution protocols have often been limited to 10 or fewer cycles of selection, but current and future technologies may enable extension far beyond this low number. It is thus illuminating to consider the relationship between the ‘fitness’ of a mutant per cycle and the number of cycles needed to raise a mutant’s frequency to detectable levels. For a discrete generation model of evolution, the form of the relationship is (5) p1=Wp0Wp0+1−p0pN=WNp0WNp0+1−p0NlnW=lnpN1−p0p01−pN,

where pt is the mutant frequency after the tth generation of selection, and W is mutant fitness (Crow & Kimura, 1970). Figure 1 shows the relationship for two combinations of initial and final frequencies; the relationship is fairly robust to differences that maintain the same ratio of pN/p0, as long as both pN and p0 are small. Even for a fitness of 2, relatively few cycles are needed to raise mutant frequency from an initial frequency of 10−7 to detectable levels (0.1).

Figure 1 Required duration of directed evolution.

Curves show relationship between mutant fitness (W) and number of cycles of selection required to raise mutant frequency from 10−7 to 0.01 (blue curve) or from 10−7 to 0.1 (red curve), from Eq. (5).

Screening may be a feasible alternative to selection

Screening is an alternative to selection, and the ready availability of flow cytometers may allow the selection of droplets based on bacterial content. If B-type cells expressed a fluorescent protein, for example, droplets could be sorted based on signal. The question with sorting is which spectrum of droplet signals to recover. For sorting of droplets, we can identify four relevant populations based on composition (Table 2). There may of course be overlap in signal between these populations, but it is reasonable to focus on composition first and then address signal.

Table 2 Categories of droplet by composition and incidence.

Category	Composition	Probability	B′ frequency	Signal	
1	C only	e−q(1 − e−χ)	–	Baseline	
2	B + C (no B′)	(1 − e−χ)(1 − e−q(1−p))e−qp	0	Low	
3	B′(±B) + C	(1 − e−χ)(1 − e−qp)	a	Medium	
4	B, B′ only	e−χ(1 − e−q)	p	High	
Notes.

Calculations based on model in Appendix; B′ frequency is after one cycle of growth; χ represents the mean number of C-type cells introduced per droplet; p is initial frequency of B′; q is mean number of B-type cells per droplet.

a =∑m=1∞∑n=0∞mm+n⋅qm+npmm!1−pnn!eqp−1eq1−p

The table shows the interesting result that only one category of droplet composition—category (3)—has a mutant frequency greater than the population frequency. The evolutionary benefit of screening depends on the ability to separate category (3) from the others, because inclusion of other categories merely dilutes the enhanced frequency of mutants in category (3) toward no mutant excess at all. However, the mutant frequency in category (3) is not simple and depends on q—the average number of B-type cells inoculated into droplets—and on the baseline mutant frequency, p. Figure 2 shows that small q and small p are the most conducive to improvement by screening. The declining benefit with large q is no doubt the simple one that the incidence of mixed droplets increases with q, and (as per our model) mixed droplets equally benefit B individuals and B′ individuals. So the advantage of B′ increasingly disappears as their benefit is increasingly shared with B.

Figure 2 Frequency of B′ cells in droplet category (3), from Table 2.

The distance above the diagonal line indicates the improvement above starting frequency obtained by screening. Thus, points in the upper left corner yield the greatest improvement from screening. There is a strong effect of q, the average number of B-type (B′ + B) cells per droplet, and a weaker effect of the population frequency of B′ (p). We are only concerned with low values of p, because once p reaches a value of even 0.1, B′ cells will be easily isolated. For low values of q, there is a profound enhancement of B′ cells, so if it is feasible to discriminate category (2) from (3), screening should be highly effective.

As noted above, the actual signal from different droplet categories will likely overlap and dilute the improvement that would be obtained from choosing category (3) alone. One especially insidious problem is that category (3) does not have the highest signal; instead, the highest signal will likely be from category (4)—droplets that were not populated with any C individuals. Thus it will not necessarily be feasible to enrich for category (3) merely by choosing droplets with the highest signal, although ensuring that nearly all droplets receive some C individuals may render category (4) to negligible levels and thereby enable screening to select just the group with highest signal. Alternatively, it may be necessary to separate fractions based on signal and search for the signal level with the highest frequency of mutants instead of the highest signal level, but that approach requires a method of recognizing favorable mutants.

Plasmid model 2: dynamics within droplets

Plasmid model 1 merely assumed final numbers of B and B′ progeny per droplet and worked out the consequences. Here we consider models of dynamics within droplets and how the biology of growth may lead to different competition outcomes. These growth models are standard differential equations assuming that bacterial growth obeys a logistic function. Indeed, the only way our model deviates from the usual assumptions of competitive growth between two species is the inclusion of a toxin (density L) produced by B that harms C. In this model, we deal only with a single B type and C type. Also, the equations neglect stochastic effects, which may be important within some droplets but not obviously over a large population of droplets. The model presented here moves us a step closer to the simulation model developed in the following section.

Parameters and variables for the model are defined in Table 3. It is assumed that the maximal growth rate of the competitor C exceeds that of B: ψC > ψB, a necessary condition for efficient selection. To accommodate the well known property that bacterial protein production declines as resources decline, the rate of toxin production is assumed proportional to the growth rate of B: (6) B˙=ψBB1−B+CKC˙=ψCC1−B+CK−γCLL˙=αB1−B+CK−δL.

Note that there are 3 parameters affecting system-wide toxin impact: the production term (α), the kill rate term (γ), and the decay rate (δ). Decay rate may depend on properties both intrinsic and extrinsic to the toxin.

Table 3 Model variables and parameters.

Notation	Definition	
Variables		
C	Density of bacteria sensitive to toxin	
B	Density of bacteria producing toxin	
L	Toxin concentration	
Parameters		
γ	Kill rate coefficient of C by toxin	
ψ C	Maximal per capita growth rate for C	
ψ B	Maximal per capita growth rate for B	
α	Toxin production rate constant	
δ	Toxin decay rate	
K	Carrying capacity of environment	

The system is at equilibrium for L = 0 and any combination of bacterial densities satisfying B + C = K. When away from equilibrium, the system ultimately moves toward this equilibrium set because B (if present) is always increasing until the combined bacterial density reaches K and because toxin concentration L ultimately goes into decline once the combined bacterial density nears K. The initial conditions thus determine the final relative abundance of B and C, but other features of the model are critical in determining the dynamics toward equilibrium. There is a rich dynamical behavior of this system; we merely offer some basic landmarks of those dynamics.

When the toxin concentration (L) is near zero, C outgrows B because its superior per capita growth rate more than compensates for the minor toxin-induced death. However, as B increases, toxin production also increases (at least while combined bacterial densities are low). If toxin density reaches the point that (7) L>ψC−ψB1−B+CKγ,

B now outgrows C, but both may nonetheless be increasing. Only when toxin density satisfies (8) ψC1−B+CK−γL<0

will C be in decline. However, the decline of C is not necessarily to extinction, because the production of L is maximal (over B) when 2B + C = K and declines to 0 at B + C = K. Thus as the combined bacterial densities increase toward K, condition (8) will reverse, allowing C to increase again.

These considerations merely hint at the possible complexity of dynamics within droplets. It is possible that an understanding of dynamics will not be essential for developing a successful protocol, but knowing details may nonetheless guide protocol improvements. The model also foreshadows the complexities of plate selection, as considered next.

Plasmid model 3: plates

The physical nature of plate competition has parallels with competition in droplets: a mix of B and C bacteria is seeded on the plate (B in the minority) and microcolonies begin growing. If a colony of B can suppress competition from nearby C, then it grows to a larger size. After a period of growth, B cells only are harvested, diluted and mixed with new C before being added to new plates. The basic selection is thus the same as before, selection favoring higher numbers of B cells.

However, there are also ways in which selection on plates differs from that in droplets, and indeed, there are phenotypes that may evolve in response to plate selection that were ignored in the droplet model. (1) There is no physical boundary for diffusion of the toxin. Thus the killing of C around a microcolony of B and competition from surviving C is continually changing as the zone of toxin expands. The model must account for toxin concentration changing over time at different distances from colonies of B. (2) Toxin molecules need not adhere (adsorb) to the first C cells encountered, so the model must include an adsorption rate. (3) Because of the first two points, it is necessary to specify the lethal effect on C cells per encounters with toxin molecules.

All of these modifications would also be included in a detailed droplet model, but the small volume and strict physical boundary of droplets obviates any need to include diffusion gradients within the droplet, at least to a suitable approximation. These additions profoundly complicate an analytical approach to plate selection, so we seek insight from simulations.

Agent-based model of plate competition

The simulations reported here used a stochastic agent-based model that has many components similar to those in the deterministic model 2 above (Eq. (6)), but having explicit spatial structure and rates that are locally determined. The population was confined to a 61 × 61 grid of patches. Each patch began with some allocation of nutrient and could be occupied by up to 3 cells. In the model, reproduction occurs at a specified rate, which depends on cell type (B or C), as long as the cell has acquired sufficient nutrient; each reproduction event costs the reproducing cell a unit of nutrient. A cell that does not have sufficient nutrient to reproduce will consume nutrient from its ‘nutrient neighborhood’ (patches that are within distance three); it will reproduce whenever it acquires 1 unit of cumulative total nutrient consumption since the last reproductive event. This is an efficient method for simulating the effects of nutrient diffusion without excessive computational cost. The offspring cell is placed at the patch of the parent or at one of the 8 neighboring patches, with pre-specified probabilities, as long as there is space available. Reproduction is suppressed whenever all these local patches are at their carrying capacity. B cells that have sufficient nutrient produce toxin molecules at a specified rate. These toxin molecules diffuse and attach to C cells, and they also degrade, all at specified rates. A C cell that acquires a specified number of toxin molecules dies and releases into the environment some amount of nutrient. This model and the agent-based phage model below were coded in NetLogo and are available in the NetLogo Modeling Commons (http://modelingcommons.org/account/login); the plasmid model is listed there under ‘Bacterial competition with toxin producer.’

From an initial seeding of B and C cells, populations were grown for 350 time steps, which was always enough to exhaust resources and stabilize cell frequencies. Each time step in the model corresponded to 3 min. To give the C cells an inherent advantage in the absence of toxin (see discussion of emulsion droplet model), we set reproduction times to 21 min (7 steps) for C and 30 min (10 steps) for B. Runs were measured for the number of B cells at this endpoint of resource exhaustion. Figures 3–5 show the amplifications (over a single simulation cycle) for different parameter values. Here ‘amplification’ is defined as B(T)/B(0), where T is the incubation time corresponding to one simulation cycle and B(t) denotes the number of B cells at time t. A ‘baseline’ value in each figure indicates the amplification in the absense of a toxin and is always plotted at x = 0.

Figure 3 B cell amplification (growth) as a function of toxin adsorption probability for different values of toxin production and toxin diffusion rate.

Toxin adsorption probability (horizontal axis) indicates the probability that a toxin molecule will attach to a C cell when they occupy the same patch; the x = 0 point corresponds to the situation with no toxin. Vertical axis (amplification) is the fold-increase in number of B cells between the beginning and end of the simulation cycle. Toxin production is the number of toxin molecules produced per simulation time step by a B cell, provided it has sufficient energy; toxin diffusion rate is the number of patches that a toxin moves (in a randomly chosen direction) each time step. The dynamic range was highest for the larger diffusion rate and corresponds to an overall 2.78× increase. The simulations were initialized with a 1 : 1 ratio of randomly distributed C to B cells (200 cells each). Other model parameters were toxin duration (the number of simulation time steps that a toxin molecule will remain viable; 3 steps), toxicity (the reciprocal of the number of attached toxin molecules needed to kill a C cell; 1/3), nutrient units per patch (12), nutrient units available per killed cell (.5), simulation time (350 steps). A step corresponds to 3 min.

Figure 4 B cell amplification as a function of toxin toxicity for different values of toxin production and toxin diffusion rate.

Toxin toxicity (horizontal axis) denotes the reciprocal of the number of attached toxin molecules needed to kill a C cell; the value x = 0 corresponds to the situation with no toxin. Toxin production is the number of toxin molecules produced per simulation time step by a B cell, provided it has sufficient energy; toxin diffusion rate is the number of patches that a toxin moves (in a randomly chosen direction) each time step. The dynamic range was highest for the larger diffusion rate and corresponds to an overall 2.67 × increase. The simulations were initialized with a 1 : 1 ratio of randomly distributed C to B cells (200 cells each). Other model parameters were toxin duration (3 steps), nutrient units per patch (12), nutrient units available per killed cell (.5), simulation time (350 steps).

Figure 5 B Cell amplification as a function of toxin duration for different initial cell densities.

Toxin duration (horizontal axis) indicates the number of simulation time steps that a toxin molecule will remain viable; the value x = 0 corresponds to the situation with no lysin. The initial numbers of cells ranged among the values 200 (1.85% of available space), 400 (3.7%), and 600 (5.5%). Clearly a judicious choice of initial conditions can make it substantially easier to select improved toxins. Toxin production is the number of toxin molecules produced per simulation time step by a B cell, provided it has sufficient energy (a value of 2 was used here); toxin diffusion rate is the number of patches that a toxin moves (in a randomly chosen direction) each time step (2); toxin adsorption probability indicates the probability that a toxin molecule will attach to a C cell when they occupy the same patch (0.33). The dynamic range was highest for lower cell densities and corresponds to an overall 2.8 × increase for one initial cell density and 4.25 × for a different initial configuration. Other model parameters were toxin toxicity (1/3), nutrient units per patch (12), nutrient units available per killed cell (.5), simulation time (350 steps).

The simulations support both the analytical results and intuition in the nature of selection under the competition protocol. For example, greater toxicity and slower toxin decay rates are favored. However, there are also interactions in the magnitudes and possibly direction of selection. Thus, progressive increases in lysin adsorption rate are strongly favored in some backgrounds but only weakly or not at all in others. Changes in multiple traits can be favored together, as can be inferred by postulating a mutation that changes both the trait displayed on the horizontal axis and one or more traits that cause an increase in the height of a curve in the graph. Absent data to suggest realistic parameter values, there is little gained by providing exhaustive iterations across parameter space. Appropriately parameterized simulations should be a good and perhaps essential augmentation of experiments, however, as they can be tailored to the biological details of the system in hand.

The amplifications illustrated may appear to be relatively modest. However, the simulations correspond to a shorter period of cell growth than would apply in a plate environment. The grid size allows at most a 28-fold expansion of population size during a simulation cycle. Growth on a plate seeded with even 106 bacteria is easily 1,000-fold. Thus the gain of a simulation cycle should at least be squared to represent a single plate, and with this correction, the most extreme improvements span 7-fold and would thus require less than 10 cycles of selection to raise mutant frequencies to detectable levels. The simulations are encouraging, therefore, in suggesting that plate selection is potentially feasible. As in the droplet model, the initial conditions of competition outcomes are important in enabling selection to increase the frequency of mutants. Finally, it is worth noting the nontrivial fact that conditions resulting in a more effective toxin also produce more B cells, as desired. This confirms the initial premise of this study, that it is possible to select for an improved toxin by conditions that improve the amplification of the toxin producer.

Phage model: droplets and plates

Alternative to encoding the toxin on a plasmid in a bacterium, the toxin gene may be encoded in a phage genome; toxin expression must of course be compatible with the intracellular life cycle of the phage in host B. However, as noted in the Introduction, a phage will not evolve to harm future hosts, as harming future hosts would translate into fewer phage descendants. Instead, the phage must evolve to increase its descendants. Thus the toxin that is intended for C must not harm B either intracellularly or extracellularly, but at the same time, the phage must encode the usual machinery to infect and kill its host, B. For lysins, a special design is therefore required, one in which the phage carries both its native lysin (against B) and a second lysin—the one to be evolved—that kills C from without. This latter lysin must not interfere with the phage life cycle in B. Whether a phage genome can tolerate dual lysins is not yet clear.

One context for selecting a phage-encoded toxin is to grow the phage on a mix of two species, one species (B) is the host for the phage (and thus produces the toxin when infected by the phage) and the other species (C) is an uninfected competitor of its host. Then the phage is selected to harm the competitor via toxin production to enhance its host’s growth and indirectly improve its own reproductive success. To carry out a sequence of competitions when the gene is carried on a phage, one would co-culture B, C, and phage and harvest the phage at the end of each incubation period. Enhanced growth of B would translate to increased phage production. Harvested phage would be pooled and then added to fresh lawns of B and C. Spatially structured growth is again required to evolve improved toxin killing by phages.

As an approximation, the key dynamic to phage growth in this process lies in the total number of B hosts generated by the end of the phage growth phase: if a phage mutant modifies the environment so that more of its hosts are produced, the mutant phage will increase (we assume that all hosts will ultimately become infected and ignore differences in host quality). However, working against this process is typically a much faster growth rate of the phage than of the bacteria.

Dynamics in droplets

Analysis of the droplet model is useful both to help establish the change in biology required by encoding the toxin on a phage as well as to illustrate a non-viable protocol. The following equations describe the dynamics in well-mixed populations of phage and bacteria (e.g., droplets), where notation is defined in Table 4: (9) P˙=−a1PB+ba1PτBτ

(10) B˙=−a1PB+ψBB1−B+CK

(11) C˙=ψCC1−B+CK−a2LC

(12) L˙=Za1PτBτ−a2LC−δL.

A variable with a subscript τ indicates the value τ minutes in the past, the time from phage infection to lysis of its host. These equations omit the spatial component that would accompany plaque growth on plates but do incorporate the temporal dimension that would apply in droplets.

Table 4 Model variables and parameters for selection of phage-endoded toxins.

Notation	Definition	
Variables		
B	Density of host bacteria	
C	Density of competitor bacteria	
P	Density of phage	
L	Density of toxin produced by phage	
Parameters		
a 1	Adsorption rate constant of phage to host bacterium (mL/min)	
a 2	Adsorption rate constant of enzyme to competitor bacterium (mL/min)	
b	Burst size of phage	
τ	Lysis time (min)	
ψ B	Growth rate of host bacterium, B (/min)	
ψ C	Growth rate of competitor bacterium, C (/min)	
Z	Enzyme production per lysis of a phage-infected cell	
δ	Decay rate of lysin	
K	Carrying capacity of environment	

The abundance of species B through time is thus affected by two processes: phage killing reduces B numbers, whereas inhibition of C (by phage toxin) increases B numbers. These two process operate at different time scales, and intuition suggests that the difference in time scales is not favorable for selecting a better toxin. Phage killing is rapid and in proportion to phage numbers. The benefit of reduced competition with C is delayed, a consequence of former B deaths that released toxin, the toxin then reducing C, and the resources that would have been used by C going toward faster growth of B. As phage reproduction is typically an order of magnitude or more faster than bacterial reproduction, and as resource limitation is unlikely to slow the growth of B until the densities near saturation (at which point phage densities are high), the benefit to the phage from this process is likely to be small.

Numerical solution of these equations using realistic parameter values for the phage-bacterial interaction support this intuition (not shown). Indeed, although changes in many of the parameter values affected the dynamics and even affected the final phage abundance, changes in the parameters affecting lysin efficacy (δ, a2) had only effects on final phage abundance when all other parameters were held constant. The model solutions thus substantiate the impression that a phage-encoded lysin will not readily be improved by directed evolution in droplets. Greater realism could be introduced by accommodating the small volume and small population sizes in droplets, but in view of the complete lack of support for lysin improvement in the current model, there seems to be little benefit of such an effort. Instead, a model of directed evolution on plates seems a more promising way to proceed.

Growth on plates may overcome some of the problems in droplets. In particular, a faster diffusion rate of lysin than of phage will create zones of dead C into which phage have not yet diffused, giving B time to grow on the resources made available by the lysin. Again, the dynamics are challenging for pure intuition, so we resort to a simulation.

Agent-based model of competition on plates

This model (‘Bacterial competition influenced by phage-encoded lysin’ in the NetLogo Modeling Commons) is similar to the previous agent-based model, with the addition of phage and associated phage parameters (lysis time, burst size, adsorption rate). Phage infect only B, and lysin is released when the infection matures and ruptures the cell. Lysin in turn diffuses and attaches to C at some rate and kills according to the number of lysins attached.

Following inoculation of the grid with B, C, and phage, populations were grown until resources were exhausted. Runs were measured for the number of phage at this endpoint. Figure 6 show the amplifications for different parameter values relative to a baseline. Here ‘amplification’ is defined as P(T)/P(0), where T is the incubation time for a cycle and P(t) denotes the number of phage at time t. The ‘baseline’ value in each figure indicates the amplification in phage in the absense of a lysin and is always plotted at x = 0.

Figure 6 Phage amplification as a function of lysin toxicity and duration.

Toxin toxicity (horizontal axis) denotes the inverse of the number of lysin molecules that must attach to a C cell in order to kill it; the value x = 0 corresponds to the situation with no lysin. Lysin duration is measured in simulation time steps (with each time step corresponding to 3 min). The two figures correspond to a phage reproduction time of 3 steps (top) and 7 steps (bottom), with 7 steps being the reproduction time of the more quickly growing C cells. The dynamic range was highest for longer lysin duration. This represented an overall 1.69 × increase (top) and a 1.89 × increase (bottom). The simulations were initialized with a 3 : 1 ratio of randomly distributed C to B cells (600 and 200, resp.) and 6 phage that were evenly spaced. Even spacing of the initial phage led to less interference between plaques and resulted in a better dynamic range and smaller error bars than when the phage are randomly located. Other model parameters were phage adsorption probability (0.25), phage burst size (20), lysin diffusion rate (1), lysin production per lysed cell (20 molecules), lysin adsorption probability (0.75), nutrient units per patch (3), nutrient units available per killed cell (1), simulation time (1,000 steps).

The simulations showed that a phage-encoded lysin can only be effective in increasing phage numbers if the lysin diffusion rate exceeds the diffusion rate of phage. In experimental systems, this condition may well be met since lysin molecules are much smaller than phage. However, diffusion dynamics are confounded by attachment rates for both lysin molecules and phage, and these are subject to evolution. For example, some phages (growing on a bacterial monoculture and not producing diffusible lysins) have been shown to evolve lower attachment rates on plates, a strategy that allows access to more susceptible hosts (Roychoudhury et al., 2014; Gallet, Kannoly & Wang, 2011).

In general, the models suggest that phage-encoded lysins have much less promise than plasmid-encoded lysins. The greatest dynamic range, approximately a 2× increase, occurred as toxicity was varied. Other parameters (lysin adsorption and diffusion rates) led to an increase of 1.3× or less. This is likely due to the fact that the phage harms B cells in addition to killing competitor C cells. Moreover, the relatively fast phage generation time does not allow much time for additional B cells to grow after lysin-induced death of C cells.

Discussion

Phage lysins show considerable promise as novel antibiotics, but their pharmacodynamic properties remain less than might be hoped. Directed evolution is a method commonly used to create and improve molecular function, but only when that improved function can be coupled with improved molecular survival or reproduction. To evolve a toxin to better kill a bacterium, directed evolution requires that the toxin gene benefit from improved killing. The context suggested in this paper is to encode the toxin in a host bacterium (e.g., on a plasmid) that exports the toxin to kill a competitor (designated the ‘plasmid’ model); the host bacteria cannot be harmed by the toxin, but the competitor must be able to greatly outgrow the host when it is not killed by toxin. Alternatively, the toxin may be encoded on a phage, killing a competitor of the host for the phage (the ‘phage’ model). Two environmental contexts were offered for enabling the necessary spatial structure: emulsion droplets and plates.

In contrast to the typical directed evolution experiment, which selects a phenotype from a baseline of zero (nothing grows unless there is improvement in the selected phenotype), directed evolution in this problem must use a baseline above zero. Here, at the outset of a cycle of selection, the host (B) must be introduced at some level, whence B must be able to maintain itself in the presence of competitor C. The most efficient baseline conditions have C growing much faster than B so that B grows to higher levels when C is suppressed than when C is not suppressed. The baseline growth of B (with no suppression of C) will thus often be well above zero, and selection can merely improve the growth of B on top of this baseline. Although these experimental conditions are challenging, they can at least be established prior to any attempts to evolve the toxin. The competitor species is known in advance—it is the species against which the toxin is directed—but there are few constraints on which host species to use for encoding the toxin.

Models assuming a plasmid-encoded toxin (whether in droplets or on plates) suggest that successful directed evolution is possible, at least if the lysin is capable of evolving improved killing. Furthermore, screening may lead to faster improvement than selection, at least in droplets, so both approaches may be attempted in parallel. The amount of progress in these protocols depends on the spectrum of mutations available (their fitness effects), on the strength of selection that can be imposed by the protocol, and on the number of cycles of selection. Directed evolution protocols have commonly been confined to relatively small numbers of cycles. Increasing the number of cycles is an obvious way to yield better outcomes.

Contrary to intuition, selection of improved toxin killing in the phage model is problematic. The challenge is that the toxin must kill the competitor species early enough that the phage host (B) can amplify its numbers before resources run out and before it is killed by the phage. When the selection was modeled as occurring in droplets, we were unable to identify conditions in which better lysins provided more than a trivial benefit to the phage. If modeled as selection on plates, lysin did evolve toward the desired goal, but the rate of evolution still fell short of that which could be obtained in the plasmid model. Evolution on plates was superior to that in droplets only when the lysin diffused faster than the phage, but fast phage growth slowed evolutionary progress even under that condition. It is of course possible that some conditions will enable adequate evolutionary progress with a phage-encoded lysin, but considerable effort may be needed to identify and then implement those conditions.

The phenotypes that respond to selection need not improve therapy. The goal of directed evolution will usually be to improve the lysin for clinical use, but gains from a directed evolution protocol need not translate into clinical benefit. Improved success of the toxin in the directed evolution protocol could stem from (i) increased toxin output, (ii) lower deleterious effect of the toxin on host B, (iii) increased diffusion of the lysin in plate selection, (iv) greater killing of C per toxin molecule, (v) increased toxin longevity, or (vi) altered binding of the toxin to C. Greater clinical success of the toxin would result from (iv) and (v) but not from (i) or (ii) and not necessarily from (iii) or (vi). In the latter case, native lysins bind with high affinity (Loessner et al., 2002; Schmelcher et al., 2010), and it seems that reduced binding can improve lysis from without, but there is likely an optimum (Cheng & Fischetti, 2007).

If progress under directed evolution does not yield clinical improvement, what should be done? An evolutionary response that does not improve the toxin for treatment might possibly be overcome simply by continuing the selection beyond any useless changes, resetting the baseline kill rate each time to favor rapid improvement in the next set of changes. Eventually, the toxin should respond in the desired phenotypes. Alternatively, the system may be engineered to suppress unwanted genetics responses to selection (e.g., designing an expression/export level that is not easily evolved to higher levels).

A direct parallel to the use of phage lysins in killing a competitor is the use of bacteriocins, molecules that have been evolved in bacteria presumably to kill competitor bacteria (Riley & Wertz, 2002; Riley et al., 2012). The models developed here for lysins apply equally to bacteriocins. However, as noted above, because naturally-occurring bacteriocins are encoded on plasmids that also carry defenses (immunity) against the bacteriocin, the selection for improved toxicity can be carried out within one species.

The models here precede published attempts at directed evolution of lysins. It may thus seem premature to develop such models. However, the nature of selection required for directed evolution of lysins is so unusual and fraught with pitfalls that the models may be a prerequisite to the development of successful protocols. The fact that lysins have been developed as therapeutics and that some modifications have been discovered by small scale screening reinforces theoretical efforts to guide their further development.

We thank Dan Nelson for advice over the years, comments on the manuscript and for providing a plasmid of plyGBS (to JJB) that was used in (unsuccessful) directed evolution protocols. Those failures in fact motivated the theoretical study presented here. Bert Baumgaertner was most generous in helping us with NetLogo. Figures were drawn in R (R Development Core Team, 2012).

Appendix: Selection when droplets are populated by more than one B-type

If droplets are initialized to have (Poisson average) q > 0 B-type cells, then PkB-types in a droplet=e−q qkk!,k=0,1,2,….

The probability of initializing a droplet with m B′ and n B cells is (A.1) PB′,B=m,n=e−qqm+nm+n!⋅m+nmp0m1−p0n,m,n≥0.

The special cases of “pure” droplets appear as (A.2) PB′,B=m,0=e−qqmm!⋅p0mandPB′,B=0,n=e−qqnn!⋅1−p0n.

Pure B′ + C droplets produce h′B′ cells and pure B + C droplets produce h B cells after incubation. For mixed droplets we assume that B and B′ cells benefit equally from the superior lysin produced by B′ cells, and hence the output of a mixed droplet is taken to be h′ total B-type cells, with these being apportioned to B′ and B according to the initial frequencies.

Thus, the average numbers of B′ and B cells after the first incubation are proportional to (A.3) h′e−q∑m=1∞∑n=0∞mm+n⋅qm+np0mm!1−p0nn!of B′

and (A.4) he−q∑n=1∞qn1−p0nn!+h′e−q∑m=1∞∑n=1∞nm+n⋅qm+np0mm!1−p0nn!(of B).

Assuming a Poisson distribution with mean q for the initializing B-types in droplets, the fraction of empty (no B-types) droplets will be e−q. For example, q = 1, 1/2 yields approximately 37% and 61% empty droplets, respectively. The probability that a droplet contains more than 2 B-type cells is 1−e−q1+q+q22.

This can be made small via the choice of q; e.g., this probability is approximately 0.08 and 0.014 for q = 1, 1/2, respectively. With this in mind, we can approximate the numbers of B′ and B cells after the first incubation; they are proportional to (A.5) h′e−qqp0+q22p02+12⋅q2p01−p0(of B′)

and (A.6) e−qhq1−p0+hq221−p02+h′12⋅q2p01−p0(of B).

In the following calculations, it is convenient to use the notation (A.7) Xg=Bg′Bg=pg1−pg,

when X0 = p0/(1 − p0). The ratio of B′ to B after one incubation cycle is thus (A.8) X1=B1′B1=h′hp01−p0⋅2+q2+q+qp0h′h−1.

These are used to begin next incubation cycle, with proportions (A.9) p1=X11+X1and1−p1=11+X1.

Iterating this procedure, we obtain the recursion in Eq. (4): (A.10) Xk+1=h′hXk⋅2+q2+q+qXk1+Xkh′h−1.

Additional Information and Declarations

Competing Interests

Author Contributions

Data deposition

The authors declare there are no competing interests.

James J. Bull and Stephen M. Krone conceived and designed the experiments, analyzed the data, wrote the paper, prepared figures and/or tables, reviewed drafts of the paper.

Cameron Crandall and Anna Rodriguez performed the experiments, analyzed the data, prepared figures and/or tables.

The following information was supplied regarding the deposition of related data:

NetLogo Modeling Commons:

http://modelingcommons.org/browse/one_model/4262#model_tabs_browse_info and http://modelingcommons.org/browse/one_model/4261#model_tabs_browse_info

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
