# Peer review of "Models for the directed evolution of bacterial allelopathy: bacteriophage lysins"

_PeerJ, doi:10.7717/peerj.879_

## Round 0.1 · original submission · Minor Revisions

Both reviewers note that that they lack expertise in mathematical modelling. Yet, they both are at the same time positive about the accessibility and clarity of the presented results. The question of the validity of a mathematical model seems to me in any case difficult to assess, because such models always have to idealize.

As per PeerJ requirements (see the penultimate bullet in section 1 of: https://peerj.com/about/policies-and-procedures/#data-materials-sharing) you should also make the source code openly available in some repository of software for scientific computing and link to it from the paper

Also the second reviewer pointed out some minor revisions, which you should take care or.

Reviewer 1 ·

Basic reporting

No comment.

Experimental design

No comment.

Validity of the findings

No comment.

Additional comments

The manuscript is well written. The complex mathematical modeling is, I must admit, a bit beyond my range of expertise in this regard, and I must largely assume it's validity in some cases. However, it is written well enough that it clarified a surprising amount of what might otherwise be a foreign language to me. My only reservation is a fundamental one, and one that I always have with this sort of investigation, and that is whether many of the assumptions will, in fact, hold true in the a real, stochastic world. When it comes to toxins, and how they influence targets (e.g. mechanisms of action, availability), there is seemingly never-ending diversity such that exceptions to this somewhat simplified description of biochemical interactions between species might ultimately fail if presented with such multifactorial "unknowns." But, again, I reckon that this is the nature of this type of modeling, and it lays a nice groundwork for subsequent experimental work.

·

Basic reporting

See "General Comments for Authors" for the complete set of comments.

Experimental design

See "General Comments for Authors" for the complete set of comments.

Validity of the findings

See "General Comments for Authors" for the complete set of comments.

Additional comments

Two caveats before the review:

First, for full transparency, my name is Daniel Nelson. I work with the class of lysin enzymes discussed in this manuscript and a few years ago I tried a directed evolution study on these enzymes with J.J. Bull, the lead author of this manuscript. While our experiments failed due to many variables we had not considered, they nonetheless provided inspiration for J.J. Bull to commence the current study developing a mathematical model of directed evolution for these enzymes with his co-authors. As such, I am listed in the acknowledgements for the motivation provided by those early failed experiments. However, at no time did I play any role in the science, writing, or editing of this manuscript and I had no prior knowledge of the manuscript contents until after the manuscript was submitted. I disclosed this information to the editorial staff prior to agreeing to review the manuscript.

Second, I am a protein biochemist/microbiologist and have 15 years of experience working on the biological aspects of these lysins. In this capacity, I can comment globally on the biology of these enzymes and some of the assumptions used by the authors to build their model. However, I am not qualified to assess the computational aspects of the manuscript. I disclosed this information to the editorial staff prior to agreeing to review the manuscript and urged them to find additional peer reviewers that do possess the computational expertise I lack.

On to the review…..

Phage Model – The foundation for the phage model needs a better explanation. In terms of evolving a generic “toxin”, the premise for the phage experiments makes sense, i.e., that some encoded toxin may harm C but not be deleterious toward B. However, in this situation, the “toxin” is a phage lysin, and if it is the lysin of the phage that infects B, then by definition it must function to lyse B in order to release progeny phage regardless of its level of “lysis from without” activity against C. Do the authors truly intend to evolve the lysin of the phage that infects B, which was my interpretation after reading the text? If so, then the statement “the toxin that is intended for C must not harm B either intracellularly or extracellularly” is not supported here. This would also add several additional layers of complexity as increasing lysis from without activity against C may likewise increase lysis from without activity against neighboring B cells (i.e. hosts for the progeny), which would presumably work against the stated directed evolution goals. An alternate method would be to incorporate a second lysin, completely unrelated to the phage and having no specificity for B, into the phage genome. In this scenario, Lysin1 is the native phage lysin that only works against B and Lysin2, the target of directed evolution, is the introduced lysin with activity against C, but not B. In this respect, the model is consistent with the generic toxin premise from above and does satisfy “the toxin that is intended for C must not harm B either intracellularly or extracellularly”. However, if this scenario is what was envisioned by the authors, then the text needs to be clarified because this setup is certainly not communicated to the reader.

Discussion – The authors list many conditions that would result in phenotypic gains from the directed evolution and correctly point out that not all of them may yield an improved lysin for clinical use. One of the gains that the authors claim will benefit improved clinical use is “(vi) increased binding of the toxin to C”. I presume the author’s rationale for making this statement relates to the low concentration of enzyme in the extracellular environment such that an increase in binding to C would presumably lead to greater killing of C. While this may be true for a generic “toxin”, I’m not sure this is supported by the literature in the case of lysins. Lysin cell wall binding domains are known to bind target cells with very tight (nanomolar) affinity (Loessner, 2002; Schmelcher, 2010). While difficult to prove, it could be speculated that this high affinity is one mechanism by which phage have evolved lysins to remain associated on cell walls of lysed host cells and not diffuse to other host cells causing lysis from without. Indeed, when lysins are engineered to contain multiple cell wall binding domains, the affinity for target cells increases dramatically but the observed lytic activity from without plummets, presumably due to a deceased catalytic efficiency (i.e. high off rate) (Schmelcher, 2011). In contrast, factors that decrease affinity of the cell wall binding domain for the bacterial surface can significantly increase the activity (turnover rate) of lysins. This can be accomplished by increasing the ionic environment by addition of NaCl (multiple manuscripts) or truncation of the cell wall binding domain (Cheng, 2007; Navaree, 1999). For lysins, there seems to be interplay between on-rates and off-rates for the bacterial peptidoglycan with phage preferring their lysins to be tight binders and scientists interested in therapeutic use of these enzymes preferring slightly lower binding resulting in a higher turnover rate. Thus, in the case of lysins, “(vi) increased binding of the toxin to C” may need to be reconsidered.

Miscellaneous comments – While the authors are correct that there have not been any formal competition-based directed evolution experiments applied to bacteriophage lysins, there have been several investigations demonstrating that a single, or few, point mutants in these enzymes can have a dramatic effect on phenotype. For example, structure-guided point mutagenesis (i.e. rational protein design) has been used to increase lysin activity (Díez-Martínez, 2013) and random mutagenesis has been used to successfully increase activity (Cheng, 2007) or increase thermostability (Heselpoth, 2012). This doesn’t affect the model in the current manuscript, but it may help provide additional biological context to the manuscript or provide a discussion point in relation to the number of directed evolution cycles required. I leave it to the authors whether they want to incorporate this additional information or not.

Acknowledgements – “plyS” was actually “PlyGBS”.

Overall, the manuscript is well written and while I cannot evaluate the computational aspects of the manuscript, the approach, variables listed, and assumptions for the models and simulations all seem logical and plausible to me. I agree with the authors that models such as these are important to help inform experimentalists on design of future directed evolution experiments with these and similar enzymes/proteins.

References cited in this review:

Cheng Q, Fischetti VA. Mutagenesis of a bacteriophage lytic enzyme PlyGBS significantly increases its antibacterial activity against group B streptococci. Appl Microbiol Biotechnol. 2007 Apr;74(6):1284-91

Díez-Martínez R, de Paz HD, Bustamante N, García E, Menéndez M, García P. Improving the lethal effect of cpl-7, a pneumococcal phage lysozyme with broad bactericidal activity, by inverting the net charge of its cell wall-binding module. Antimicrob Agents Chemother. 2013 Nov;57(11):5355-65.

Heselpoth RD, Nelson DC. A new screening method for the directed evolution of thermostable bacteriolytic enzymes. J Vis Exp. 2012 Nov 7;(69). pii: 4216.

Loessner MJ, Kramer K, Ebel F, Scherer S. C-terminal domains of Listeria monocytogenes bacteriophage murein hydrolases determine specific recognition and high-affinity binding to bacterial cell wall carbohydrates. Mol Microbiol. 2002 Apr;44(2):335-49.

Navarre WW, Ton-That H, Faull KF, Schneewind O. Multiple enzymatic activities of the murein hydrolase from staphylococcal phage phi11. Identification of a D-alanyl-glycine endopeptidase activity. J Biol Chem. 1999 May 28;274(22):15847-56.

Schmelcher M, Shabarova T, Eugster MR, Eichenseher F, Tchang VS, Banz M, Loessner MJ. Rapid multiplex detection and differentiation of Listeria cells by use of fluorescent phage endolysin cell wall binding domains. Appl Environ Microbiol. 2010 Sep;76(17):5745-56.

Schmelcher M, Tchang VS, Loessner MJ. Domain shuffling and module engineering of Listeria phage endolysins for enhanced lytic activity and binding affinity. Microb Biotechnol. 2011 Sep;4(5):651-62.

---

## Round 0.2 · accepted · Accept

I'm pleased to see the changes.